# Structured Learning via Logistic Regression

**Justin Domke**
NICTA and The Australian National University
`justin.domke@nicta.com.au`

## Abstract

A successful approach to structured learning is to write the learning objective as a joint function of linear parameters and inference messages, and iterate between updates to each. This paper observes that if the inference problem is "smoothed" through the addition of entropy terms, for fixed messages, the learning objective reduces to a traditional (non-structured) logistic regression problem with respect to parameters. In these logistic regression problems, each training example has a bias term determined by the current set of messages. Based on this insight, the structured energy function can be extended from linear factors to any function class where an "oracle" exists to minimize a logistic loss.

## 1 Introduction

The structured learning problem is to find a function $F(x, y)$ to map from inputs $x$ to outputs as $y^* = \arg\max_y F(x, y)$. $F$ is chosen to optimize a loss function defined on these outputs. A major challenge is that evaluating the loss for a given function $F$ requires solving the inference optimization to find the highest-scoring output $y$ for each exemplar, which is NP-hard in general. A standard solution to this is to write the loss function using an LP-relaxation of the inference problem, meaning an upper-bound on the true loss. The learning problem can then be phrased as a joint optimization of parameters and inference variables, which can be solved, e.g., by alternating message-passing updates to inference variables with gradient descent updates to parameters [16, 9].

Previous work has mostly focused on linear energy functions $F(x, y) = w^T \Phi(x, y)$, where a vector of weights $w$ is adjusted in learning, and $\Phi(x, y) = \sum_\alpha \Phi(x, y_\alpha)$ decomposes over subsets of variables $y_\alpha$. While linear weights are often useful in practice [23, 16, 9, 3, 17, 12, 5], it is also common to make use of non-linear classifiers. This is typically done by training a classifier (e.g. ensembles of trees [20, 8, 25, 13, 24, 18, 19] or multi-layer perceptrons [10, 21]) to predict each variable independently. Linear edge interaction weights are then learned, with unary classifiers either held fixed [20, 8, 25, 13, 24, 10] or used essentially as "features" with linear weights re-adjusted [18].

This paper allows the more general form $F(x, y) = \sum_\alpha f_\alpha(x, y_\alpha)$. The learning problem is to select $f_\alpha$ from some set of functions $\mathcal{F}_\alpha$. Here, following previous work [15], we add entropy smoothing to the LP-relaxation of the inference problem. Again, this leads to phrasing the learning problem as a joint optimization of learning and inference variables, alternating between message-passing updates to inference variables and optimization of the functions $f_\alpha$. The major result is that minimization of the loss over $f_\alpha \in \mathcal{F}_\alpha$ can be re-formulated as a logistic regression problem, with a "bias" vector added to each example reflecting the current messages incoming to factor $\alpha$. No assumptions are needed on the sets of functions $\mathcal{F}_\alpha$, beyond assuming that an algorithm exists to optimize the logistic loss on a given dataset over all $f_\alpha \in \mathcal{F}_\alpha$

We experimentally test the results of varying $\mathcal{F}_\alpha$ to be the set of linear functions, multi-layer perceptrons, or boosted decision trees. Results verify the benefits of training flexible function classes in terms of joint prediction accuracy.

## 2    Structured Prediction

The structured prediction problem can be written as seeking a function $h$ that will predict an output $y$ from an input $x$. Most commonly, it can be written in the form

$$h(x; w) = \arg\max_y w^T \Phi(x, y), \qquad (1)$$

where $\Phi$ is a fixed function of both $x$ and $y$. The maximum takes place over all configurations of the discrete vector $y$. It is further assumed that $\Phi$ decomposes into a sum of functions evaluated over subsets of variables $y_\alpha$ as

$$\Phi(x, y) = \sum_\alpha \Phi_\alpha(x, y_\alpha).$$

The learning problem is to adjust set of linear weights $w$. This paper considers the structured learning problem in a more general setting, directly handling nonlinear function classes. We generalize the function $h$ to

$$h(x; F) = \arg\max_y F(x, y),$$

where the energy $F$ again decomposes as

$$F(x, y) = \sum_\alpha f_\alpha(x, y_\alpha).$$

The learning problem now becomes to select $\{f_\alpha \in \mathcal{F}_\alpha\}$ for some set of functions $\mathcal{F}_\alpha$. This reduces to the previous case when $f_\alpha(x, y_\alpha) = w^T \Phi_\alpha(x, y_\alpha)$ is a linear function. Here, we do not make any assumption on the class of functions $\mathcal{F}_\alpha$ other than assuming that there exists an algorithm to find the best function $f_\alpha \in \mathcal{F}_\alpha$ in terms of the logistic regression loss (Section 6).

## 3    Loss Functions

Given a dataset $(x^1, y^1), ..., (x^N, y^N)$, we wish to select the energy $F$ to minimize the empirical risk

$$R(F) = \sum_k l(x^k, y^k; F), \qquad (2)$$

for some loss function $l$. Absent computational concerns, a standard choice would be the slack-rescaled loss [22]

$$l_0(x^k, y^k; F) = \max_y F(x^k, y) - F(x^k, y^k) + \Delta(y^k, y), \qquad (3)$$

where $\Delta(y^k, y)$ is some measure of discrepancy. We assume that $\Delta$ is a function that decomposes over $\alpha$, (i.e. that $\Delta(y^k, y) = \sum_\alpha \Delta_\alpha(y^k_\alpha, y_\alpha)$). Our experiments use the Hamming distance.

In Eq. 3, the maximum ranges over all possible discrete labelings $y$, which is in NP-hard in general. If this inference problem must be solved approximately, there is strong motivation [6] for using relaxations of the maximization in Eq. 1, since this yields an upper-bound on the loss. A common solution [16, 14, 6] is to use a linear relaxation[1]

$$l_1(x^k, y^k; F) = \max_{\mu \in \mathcal{M}} F(x^k, \mu) - F(x^k, y^k) + \Delta(y^k, \mu), \qquad (4)$$

where the local polytope $\mathcal{M}$ is defined as the set of local pseudomarginals that are normalized, and agree when marginalized over other neighboring regions,

$$\mathcal{M} = \{\mu | \mu_{\alpha\beta}(y_\beta) = \mu_\beta(y_\beta) \,\forall \beta \subset \alpha, \quad \sum_{y_\alpha} \mu_\alpha(y_\alpha) = 1 \,\forall \alpha, \quad \mu_\alpha(y_\alpha) \ge 1 \,\forall \alpha, y_\alpha \}.$$

Here, $\mu_{\alpha\beta}(y_\beta) = \sum_{y_{\alpha\backslash\beta}} \mu_\alpha(y_\alpha)$ is $\mu_\alpha$ marginalized out over some region $\beta$ contained in $\alpha$. It is easy to show that $l_1 \ge l_0$, since the two would be equivalent if $\mu$ were restricted to binary values, and hence the maximization in $l_1$ takes place over a larger set [6]. We also define

$$\theta^k_F(y_\alpha) = f_\alpha(x^k, y_\alpha) + \Delta_\alpha(y^k_\alpha, y_\alpha), \qquad (5)$$

which gives the equivalent representation of $l_1$ as $l_1(x^k, y^k; F) = -F(x^k, y^k) + \max_{\mu \in \mathcal{M}} \theta_F^k \cdot \mu$.

The maximization in $l_1$ is of a linear objective under linear constraints, and is thus a linear program (LP), solvable in polynomial time using a generic LP solver. In practice, however, it is preferable to use custom solvers based on message-passing that exploit the sparsity of the problem.

Here, we make a further approximation to the loss, replacing the inference problem of $\max_{\mu \in \mathcal{M}} \theta \cdot \mu$ with the "smoothed" problem $\max_{\mu \in \mathcal{M}} \theta \cdot \mu + \epsilon \sum_\alpha H(\mu_\alpha)$, where $H(\mu_\alpha)$ is the entropy of the marginals $\mu_\alpha$. This approximation has been considered by Meshi et al. [15] who show that local message-passing can have a guaranteed convergence rate, and by Hazan and Urtasun [9] who use it for learning. The relaxed loss is

$$l(x^k, y^k; F) = -F(x^k, y^k) + \max_{\mu \in \mathcal{M}} \left( \theta_F^k \cdot \mu + \epsilon \sum_\alpha H(\mu_\alpha) \right). \tag{6}$$

Since the entropy is positive, this is clearly a further upper-bound on the "unsmoothed" loss, i.e. $l_1 \leq l$. Moreover, we can bound the looseness of this approximation as in the following theorem, proved in the appendix. A similar result was previously given [15] bounding the difference of the objective obtained by inference with and without entropy smoothing.

**Theorem 1.** *$l$ and $l_1$ are bounded by (where $|y_\alpha|$ is the number of configurations of $y_\alpha$)*

$$l_1(x, y, F) \leq l(x, y, F) \leq l_1(x, y, F) + \epsilon H_{\max}, \quad H_{\max} = \sum_\alpha \log |y_\alpha|.$$

## 4   Overview

Now, the learning problem is to select the functions $f_\alpha$ composing $F$ to minimize $R$ as defined in Eq. 2. The major challenge is that evaluating $R(F)$ requires performing inference. Specifically, if we define

$$A(\theta) = \max_{\mu \in \mathcal{M}} \theta \cdot \mu + \epsilon \sum_\alpha H(\mu_\alpha), \tag{7}$$

then we have that

$$\min_F R(F) = \min_F \sum_k \left( -F(x^k, y^k) + A(\theta_F^k) \right).$$

Since $A(\theta)$ contains a maximization, this is a saddle-point problem. Inspired by previous work [16, 9], our solution (Section 5) is to introduce a vector of "messages" $\lambda$ to write $A$ in the dual form

$$A(\theta) = \min_\lambda A(\lambda, \theta),$$

which leads to phrasing learning as the joint minimization

$$\min_F \min_{\{\lambda^k\}} \sum_k \left[ -F(x^k, y^k) + A(\lambda^k, \theta_F^k) \right].$$

We propose to solve this through an alternating optimization of $F$ and $\{\lambda^k\}$. For fixed $F$, message-passing can be used to perform coordinate ascent updates to all the messages $\lambda^k$ (Section 5). These updates are trivially parallelized with respect to $k$. However, the problem remains, for fixed messages, how to optimize the functions $f_\alpha$ composing $F$. Section 7 observes that this problem can be re-formulated into a (non-structured) logistic regression problem, with "bias" terms added to each example that reflect the current messages into factor $\alpha$.

## 5   Inference

In order to evaluate the loss, it is necessary to solve the maximization in Eq. 6. For a given $\theta$, consider doing inference over $\mu$, that is, in solving the maximization in Eq. 7. Standard Lagrangian duality theory gives the following dual representation for $A(\theta)$ in terms of "messages" $\lambda_\alpha(x_\beta)$ from a region $\alpha$ to a subregion $\beta \subset \alpha$, a variant of the representation of Heskes [11].

**Algorithm 1** Reducing structured learning to logistic regression.

For all $k$, $\alpha$, initialize $\lambda^k(y_\alpha) \leftarrow 0$.

Repeat until convergence:

1. For all $k$, for all $\alpha$, set the bias term to

$$b_\alpha^k(y_\alpha) \leftarrow \frac{1}{\epsilon}\left(\Delta(y_\alpha^k, y_\alpha) + \sum_{\beta \subset \alpha} \lambda_\alpha^k(y_\beta) - \sum_{\gamma \supset \alpha} \lambda_\gamma^k(y_\alpha)\right).$$

2. For all $\alpha$, solve the logistic regression problem

$$f_\alpha \leftarrow \arg\max_{f_\alpha \in \mathcal{F}_\alpha} \sum_{k=1}^{K}\left[\left(f_\alpha(x^k, y_\alpha^k) + b_\alpha^k(y_\alpha^k)\right) - \log\sum_{y_\alpha}\exp\left(f_\alpha(x^k, y_\alpha) + b_\alpha^k(y_\alpha)\right)\right].$$

3. For all $k$, for all $\alpha$, form updated parameters as

$$\theta^k(y_\alpha) \leftarrow \epsilon f_\alpha(x^k, y_\alpha) + \Delta(y_\alpha^k, y_\alpha).$$

4. For all $k$, perform a fixed number of message-passing iterations to update $\lambda^k$ using $\theta^k$. (Eq. 10)

---

**Theorem 2.** $A(\theta)$ can be represented in the dual form $A(\theta) = \min_\lambda A(\lambda, \theta)$, where

$$A(\lambda, \theta) = \max_{\mu \in \mathcal{N}} \theta \cdot \mu + \epsilon \sum_\alpha H(\mu_\alpha) + \sum_\alpha \sum_{\beta \subset \alpha} \sum_{x_\beta} \lambda_\alpha(x_\beta)\left(\mu_{\alpha\beta}(y_\beta) - \mu_\beta(y_\beta)\right), \quad (8)$$

and $\mathcal{N} = \{\mu | \sum_{y_\alpha} \mu_\alpha(y_\alpha) = 1, \mu_\alpha(y_\alpha) \geq 0\}$ is the set of locally normalized pseudomarginals. Moreover, for a fixed $\lambda$, the maximizing $\mu$ is given by

$$\mu_\alpha(y_\alpha) = \frac{1}{Z_\alpha}\exp\left(\frac{1}{\epsilon}\left(\theta(y_\alpha) + \sum_{\beta \subset \alpha}\lambda_\alpha(y_\beta) - \sum_{\gamma \supset \alpha}\lambda_\gamma(y_\alpha)\right)\right), \quad (9)$$

where $Z_\alpha$ is a normalizing constant to ensure that $\sum_{y_\alpha} \mu_\alpha(y_\alpha) = 1$.

Thus, for any set of messages $\lambda$, there is an easily-evaluated upper-bound $A(\lambda, \theta) \geq A(\theta)$, and when $A(\lambda, \theta)$ is minimized with respect to $\lambda$, this bound is tight. The standard approach to performing the minimization over $\lambda$ is essentially block-coordinate descent. There are variants, depending on the size of the "block" that is updated. In our experiments, we use blocks consisting of the set of all messages $\lambda_\alpha(y_\nu)$ for all regions $\alpha$ containing $\nu$. When the graph only contains regions for single variables and pairs, this is a "star update" of all the messages from pairs that contain a variable $i$. It can be shown [11, 15] that the update is

$$\lambda'_\alpha(y_\nu) \leftarrow \lambda_\alpha(y_\nu) + \frac{\epsilon}{1 + N_\nu}\left(\log\mu_\nu(y_\nu) + \sum_{\alpha' \supset \nu}\log\mu_{\alpha'}(y_\nu)\right) - \epsilon\log\mu_\alpha(y_\nu), \quad (10)$$

for all $\alpha \supset \nu$, where $N_\nu = |\{\alpha | \alpha \supset \nu\}|$. Meshi et al. [15] show that with greedy or randomized selection of blocks to update, $\mathcal{O}(\frac{1}{\delta})$ iterations are sufficient to converge within error $\delta$.

## 6 Logistic Regression

Logistic regression is traditionally understood as defining a conditional distribution $p(y|x; W) = \exp\left((Wx)_y\right)/Z(x)$ where $W$ is a matrix that maps the input features $x$ to a vector of margins $Wx$. It is easy to show that the maximum conditional likelihood training problem $\max_W \sum_k \log p(y^k|x^k; W)$ is equivalent to

$$\max_W \sum_k \left[(Wx^k)_{y^k} - \log\sum_y \exp(Wx^k)_y\right].$$

Here, we generalize this in two ways. First, rather than taking the mapping from features $x$ to the margin for label $y$ as the $y$-th component of $Wx$, we take it as $f(x, y)$ for some function $f$ in a set of function $\mathcal{F}$. (This reduces to the linear case when $f(x, y) = (Wx)_y$.) Secondly, we assume that there is a pre-determined "bias" vector $b^k$ associated with each training example. This yields the learning problem

$$\max_{f \in \mathcal{F}} \sum_k \left[ \left( f(x^k, y^k) + b^k(y^k) \right) - \log \sum_y \exp \left( f(x^k, y) + b^k(y) \right) \right], \tag{11}$$

Aside from linear logistic regression, one can see decision trees, multi-layer perceptrons, and boosted ensembles under an appropriate loss as solving Eq. 11 for different sets of functions $\mathcal{F}$ (albeit possibly to a local maximum).

## 7 Training

Recall that the learning problem is to select the functions $f_\alpha \in \mathcal{F}_\alpha$ so as to minimize the empirical risk $R(F) = \sum_k [-F(x^k, y^k) + A(\theta_F^k)]$. At first blush, this appears challenging, since evaluating $A(\theta)$ requires solving a message-passing optimization. However, we can use the dual representation of $A$ from Theorem 2 to represent $\min_F R(F)$ in the form

$$\min_F \min_{\{\lambda^k\}} \sum_k \left[ -F(x^k, y^k) + A(\lambda^k, \theta_F^k) \right]. \tag{12}$$

To optimize Eq. 12, we alternating between optimization of messages $\{\lambda^k\}$ and energy functions $\{f_\alpha\}$. Optimization with respect to $\lambda^k$ for fixed $F$ decomposes into minimizing $A(\lambda^k, \theta_F^k)$ independently for each $y^k$, which can be done by running message-passing updates as in Section 5 using the parameter vector $\theta_F^k$. Thus, the rest of this section is concerned with how to optimize with respect to $F$ for fixed messages. Below, we will use a slight generalization of a standard result [1, p. 93].

**Lemma 3.** *The conjugate of the entropy is the "log-sum-exp" function. Formally,*

$$\max_{x: x^T 1 = 1, x \geq 0} \theta \cdot x - \rho \sum_i x_i \log x_i = \rho \log \sum_i \exp \frac{\theta_i}{\rho}.$$

**Theorem 4.** *If $f_\alpha^*$ is the minimizer of Eq 12 for fixed messages $\lambda$, then*

$$f_\alpha^* = \epsilon \arg\max_{f_\alpha} \sum_k \left[ \left( f_\alpha(x^k, y_\alpha^k) + b_\alpha^k(y_\alpha^k) \right) - \log \sum_{y_\alpha} \exp \left( f_\alpha(x^k, y_\alpha) + b_\alpha^k(y_\alpha) \right) \right], \tag{13}$$

*where the set of biases are defined as*

$$b_\alpha^k(y_\alpha) = \frac{1}{\epsilon} \left( \Delta(y_\alpha^k, y_\alpha) + \sum_{\beta \subset \alpha} \lambda_\alpha(y_\beta) - \sum_{\gamma \supset \alpha} \lambda_\gamma(y_\alpha) \right). \tag{14}$$

*Proof.* Substituting $A(\lambda, \theta)$ from Eq. 8 and $\theta^k$ from Eq. 5 gives that

$$A(\lambda^k, \theta_F^k) = \max_{\mu \in \mathcal{N}} \sum_\alpha \sum_{y_\alpha} \left( f_\alpha(x^k, y_\alpha) + \Delta_\alpha(y_\alpha^k, y_\alpha) \right) \mu(y_\alpha) + \epsilon \sum_\alpha H(\mu_\alpha)$$

$$+ \sum_\alpha \sum_{\beta \subset \alpha} \sum_{x_\beta} \lambda_\alpha^k(x_\beta) \left( \mu_{\alpha\beta}(y_\beta) - \mu_\beta(y_\beta) \right).$$

Using the definition of $b^k$ from Eq. 14 above, this simplifies into

$$A(\lambda^k, \theta_F^k) = \sum_\alpha \max_{\mu_\alpha \in \mathcal{N}_\alpha} \left( \sum_{y_\alpha} \left( f_\alpha(x, y_\alpha) + \epsilon b_\alpha(y_\alpha) \right) \mu_\alpha(y_\alpha) + \epsilon H(\mu_\alpha) \right),$$

| | Denoising | | | | | | Horses | | | | |
|---|---|---|---|---|---|---|---|---|---|---|---|
| $\mathcal{F}_i \setminus \mathcal{F}_{ij}$ | Zero | Const. | Linear | Boost. | MLP | $\mathcal{F}_i \setminus \mathcal{F}_{ij}$ | Zero | Const. | Linear | Boost. | MLP |
| Zero | .502 | .502 | .502 | .511 | .502 | Zero | .246 | .246 | .247 | .244 | .245 |
| Const. | .502 | .502 | .502 | .510 | .502 | Const. | .246 | .246 | .247 | .244 | .245 |
| Linear | .444 | .077 | .059 | .049 | .034 | Linear | .185 | .185 | .168 | .154 | .156 |
| Boost. | .444 | .034 | .015 | .009 | .007 | Boost. | .103 | .098 | .092 | .084 | .086 |
| MLP | .445 | .032 | .015 | .009 | .008 | MLP | .096 | .094 | .087 | .080 | .081 |

Table 1: Univariate Test Error Rates (Train Errors in Appendix)

where $\mathcal{N}_\alpha = \{\mu_\alpha | \sum_{y_\alpha} \mu_\alpha(y_\alpha) = 1, \mu_\alpha(y_\alpha) \geq 0\}$ enforces that $\mu_\alpha$ is a locally normalized set of marginals. Applying Lemma 3 to the inner maximization gives the closed-form expression

$$A(\lambda^k, \theta_F^k) = \sum_\alpha \epsilon \log \sum_{y_\alpha} \exp\left(\frac{1}{\epsilon} f_\alpha(x, y_\alpha) + b_\alpha(y_\alpha)\right).$$

Thus, minimizing Eq. 12 with respect to $F$ is equivalent to finding (for all $\alpha$)

$$\arg\max_{f_\alpha} \sum_k \left[ f_\alpha(x^k, y_\alpha^k) - \epsilon \log \sum_{y_\alpha} \exp\left(\frac{1}{\epsilon} f_\alpha(x, y_\alpha) + b_\alpha^k(y_\alpha)\right) \right]$$

$$= \arg\max_{f_\alpha} \sum_k \left[ \frac{1}{\epsilon} f_\alpha(x^k, y_\alpha^k) - \log \sum_{y_\alpha} \exp\left(\frac{1}{\epsilon} f(x^k, y_\alpha) + b_\alpha^k(y_\alpha)\right) \right]$$

Observing that adding a bias term doesn't change the maximizing $f_\alpha$, and using the fact that $\arg\max g(\frac{1}{\epsilon} \cdot) = \epsilon \arg\max g(\cdot)$ gives the result. $\qquad\square$

The final learning algorithm is summarized as Alg. 1. Sometimes, the local classifier $f_\alpha$ will depend on the input $x$ only through some "local features" $\phi_\alpha$. The above framework accomodates this situation if the set $\mathcal{F}_\alpha$ is considered to select these local features.

In practice, one will often wish to constrain that some of the functions $f_\alpha$ are the same. This is done by taking the sum in Eq. 13 not just over all data $k$, but also over all factors $\alpha$ that should be so constrained. For example, it is common to model image segmentation problems using a 4-connected grid with an energy like $F(x, y) = \sum_i u(\phi_i, y_i) + \sum_{ij} v(\phi_{ij}, y_i, y_j)$, where $\phi_i/\phi_{ij}$ are univariate/pairwise features determined by $x$, and $u$ and $v$ are functions mapping local features to local energies. In this case, $u$ would be selected to maximize $\sum_k \sum_i \left[ (u(\phi_i^k, y_i^k) + b_i^k(y_i^k)) - \log \sum_{y_i} \exp\left(u(\phi_i^k, y_i) + b_i^k(y_i)\right) \right]$, and analogous expression exists for $v$. This is the framework used in the following experiments.

## 8 Experiments

These experiments consider three different function classes: linear, boosted decision trees, and multi-layer perceptrons. To maximize Eq. 11 under linear functions $f(x, y) = (Wx)_y$, we simply compute the gradient with respect to $W$ and use batch L-BFGS. For a multi-layer perceptron, we fit the function $f(x, y) = (W\sigma(Ux))_y$ using stochastic gradient descent with momentum[2] on mini-batches of size 1000, using a step size of .25 for univariate classifiers and .05 for pairwise. Boosted decision trees use stochastic gradient boosting [7]: the gradient of the logistic loss is computed for each exemplar, and a regression tree is induced to fit this (one tree for each class). To control overfitting, each leaf node must contain at least 5% of the data. Then, an optimization adjusts the values of leaf nodes to optimize the logistic loss. Finally, the tree values are multiplied by

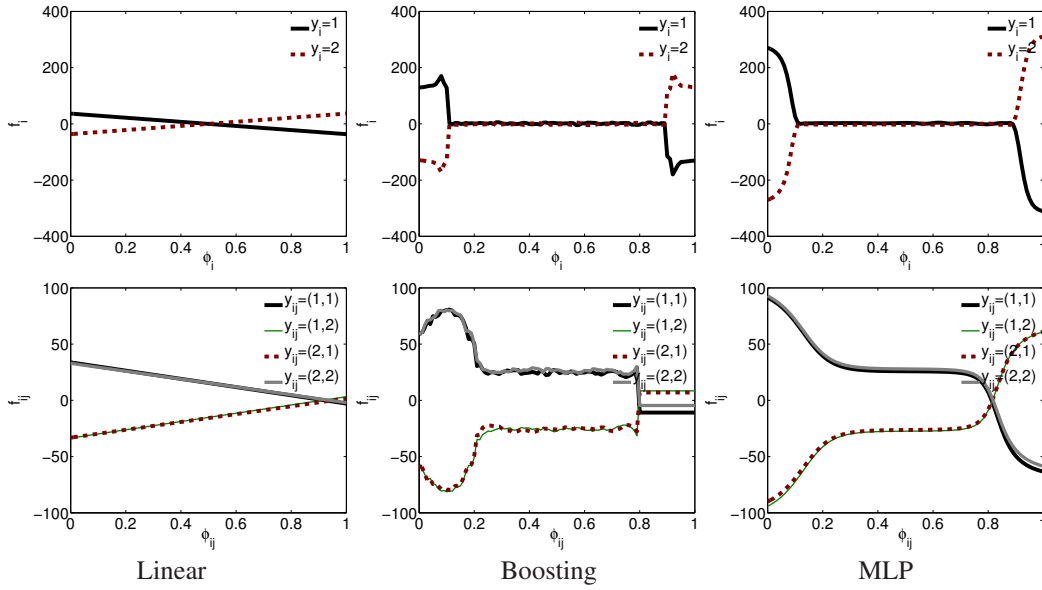

Figure 1: The univariate (top) and pairwise (bottom) energy functions learned on denoising data. Each column shows the result of training both univariate and pairwise terms with one function class.

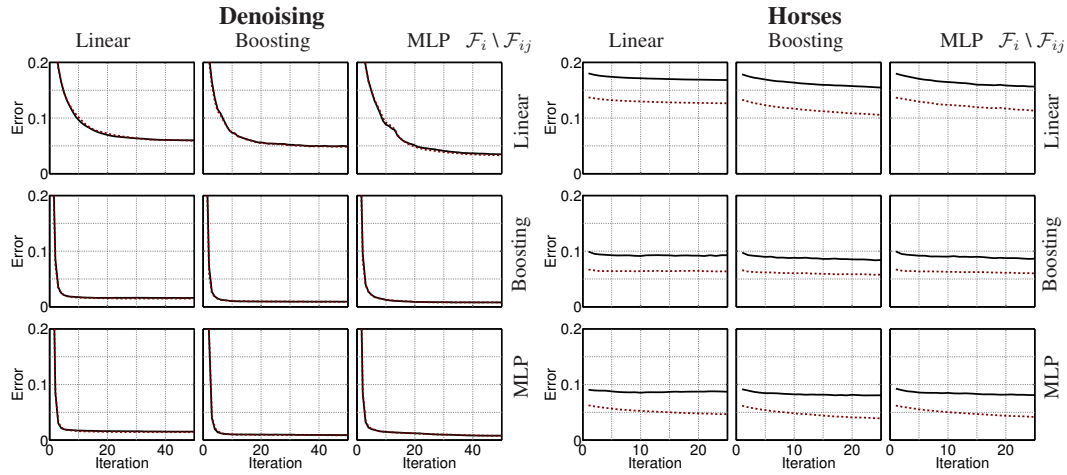

Figure 2: Dashed/Solid lines show univariate train/test error rates as a function of learning iterations for varying univariate (rows) and pairwise (columns) classifiers.

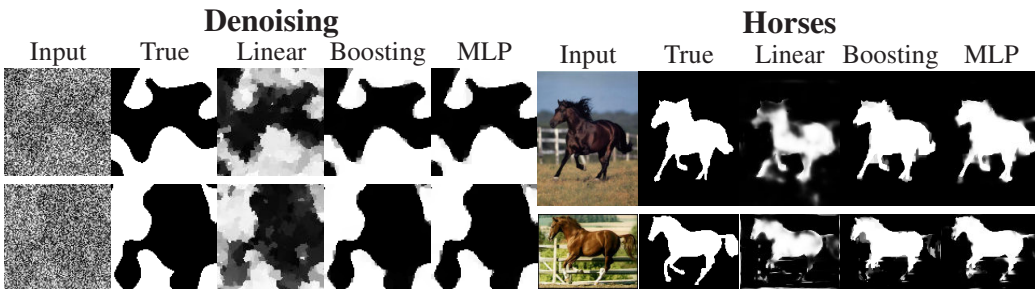

Figure 3: Example Predictions on Test Images (More in Appendix)

.25 and added to the ensemble. For reference, we also consider the "zero" classifier, and a "constant" classifier that ignores the input– equivalent to a linear classifier with a single constant feature.

All examples use $\epsilon = 0.1$. Each learning iteration consists of updating $f_i$, performing 25 iterations of message passing, updating $f_{ij}$ , and then performing another 25 iterations of message-passing.

The first dataset is a synthetic binary denoising dataset, intended for the purpose of visualization. To create an example, an image is generated with each pixel random in $[0, 1]$. To generate $y$, this image is convolved with a Gaussian with standard deviation 10 and rounded to $\{0, 1\}$. Next, if $y_i = 0$, $\phi_i^k$ is sampled uniformly from $[0, .9]$, while if $y_i^k = 1$, $\phi_i^k$ is sampled from $[.1, 1]$. Finally, for a pair $(i, j)$, if $y_i^k = y_j^k$, then $\phi_{ij}^k$ is sampled from $[0, .8]$ while if $y_i^k \neq y_j^k$ $\phi_{ij}$ is sampled from $[.2, 1]$. A constant feature is also added to both $\phi_i^k$ and $\phi_{ij}^k$.

There are 16 $100 \times 100$ images each training and testing. Test errors for each classifier combination are in Table 1, learning curves are in Fig. 2, and example results in Fig. 3. The nonlinear classifiers result in both lower asymptotic training and testing errors and faster convergence rates. Boosting converges particularly quickly. Finally, because there is only a single input feature for univariate and pairwise terms, the resulting functions are plotted in Fig. 1.

Second, as a more realistic example, we use the Weizmann horses dataset. We use 42 univariate features $f_i^k$ consisting of a constant (1) the RBG values of the pixel (3), the vertical and horizontal position (2) and a histogram of gradients [2] (36). There are three edge features, consisting of a constant, the $l_2$ distance of the RBG vectors for the two pixels, and the output of a Sobel edge filter. Results are show in Table 1 and Figures 2 and 3. Again, we see benefits in using nonlinear classifiers, both in convergence rate and asymptotic error.

## 9    Discussion

This paper observes that in the structured learning setting, the optimization with respect to energy can be formulated as a logistic regression problem for each factor, "biased" by the current messages. Thus, it is possible to use any function class where an "oracle" exists to optimize a logistic loss. Besides the possibility of using more general classes of energies, another advantage of the proposed method is the "software engineering" benefit of having the algorithm for fitting the energy modularized from the rest of the learning procedure. The ability to easily define new energy functions for individual problems could have practical impact.

Future work could consider convergence rates of the overall learning optimization, systematically investigate the choice of $\epsilon$, or consider more general entropy approximations, such as the Bethe approximation used with loopy belief propagation.

In related work, Hazan and Urtasun [9] use a linear energy, and alternate between updating all inference variables and a gradient descent update to parameters, using an entropy-smoothed inference objective. Meshi et al. [16] also use a linear energy, with a stochastic algorithm updating inference variables and taking a stochastic gradient step on parameters for one exemplar at a time, with a pure LP-relaxation of inference. The proposed method iterates between updating all inference variables and performing a full optimization of the energy. This is a "batch" algorithm in the sense of making repeated passes over the data, and so is expected to be slower than an online method for large datasets. In practice, however, inference is easily parallelized over the data, and the majority of computational time is spent in the logistic regression subproblems. A stochastic solver can easily be used for these, as was done for MLPs above, giving a partially stochastic learning method.

Another related work is Gradient Tree Boosting [4] in which to train a CRF, the functional gradient of the conditional likelihood is computed, and a regression tree is induced. This is iterated to produce an ensemble. The main limitation is the assumption that inference can be solved exactly. It appears possible to extend this to inexact inference, where the tree is induced to improve a dual bound, but this has not been done so far. Experimentally, however, simply inducing a tree on the loss gradient leads to much slower learning if the leaf nodes are not modified to optimize the logistic loss. Thus, it is likely that such a strategy would still benefit from using the logistic regression reformulation.

## Footnotes

[1]Here, $F$ and $\Delta$ are slightly generalized to allow arguments of pseudomarginals, as $F(x^k, \mu) = \sum_\alpha \sum_{y_\alpha} f(x^k, y_\alpha)\mu(y_\alpha)$ and $\Delta(y^k, \mu) = \sum_\alpha \sum_{y_\alpha} \Delta_\alpha(y^k_\alpha, y_\alpha)\mu(y_\alpha)$.

[2]At each time, the new step is a combination of .1 times the new gradient plus .9 times the old step.

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
