[Supplementary Material · StructuredLearningPerspective8_appendixonly.pdf]

# Appendix for paper: Structured Learning via Logistic Regression

**Theorem 5.** *The difference of $l$ and $l_1$ is bounded by*

$$l_1(x, y, F) \leq l(x, y, F) \leq l_1(x, y, F) + \epsilon H_{\max}, \quad H_{\max} = \sum_\alpha \log |y_\alpha|.$$

*Proof.* Defining $\mu^* = \arg\max_{\mu \in \mathcal{M}} \theta \cdot \mu$ and $\mu' = \arg\max_{\mu \in \mathcal{M}} \theta \cdot \mu + \epsilon \sum_\alpha H(\mu_\alpha)$, one can write

$$
\begin{aligned}
l(x, y; F) - l_1(x, y; F) &= -F(x, y) + \max_{\mu \in \mathcal{M}} \left( \theta \cdot \mu + \sum_\alpha \epsilon H(\mu_\alpha) \right) + F(x, y) - \max_{\mu \in \mathcal{M}} \theta \cdot \mu \\
&= \max_{\mu \in \mathcal{M}} \left( \theta \cdot \mu + \sum_\alpha \epsilon H(\mu_\alpha) \right) - \max_{\mu \in \mathcal{M}} \theta \cdot \mu \\
&= \theta \cdot \mu' - \theta \cdot \mu^* + \sum_\alpha \epsilon H(\mu'_\alpha) \\
&\leq \epsilon \sum_\alpha \log |y_\alpha|.
\end{aligned}
$$

The last line follows from the fact that $\theta \cdot \mu^* \geq \theta \cdot \mu'$, and that $H(\mu'_\alpha) \leq \log |y_\alpha|$. $\qquad\square$

**Denoising**

| $\mathcal{F}_i \setminus \mathcal{F}_{ij}$ | Zero | Const. | Linear | Boost. | MLP |
|---|---|---|---|---|---|
| Zero | .490 | .490 | .490 | .441 | .490 |
| Const. | .490 | .490 | .490 | .440 | .490 |
| Linear | .443 | .077 | .059 | .048 | .033 |
| Boost. | .429 | .032 | .014 | .008 | .008 |
| MLP | .435 | .031 | .014 | .008 | .008 |

**Horses**

| $\mathcal{F}_i \setminus \mathcal{F}_{ij}$ | Zero | Const. | Linear | Boost. | MLP |
|---|---|---|---|---|---|
| Zero | .211 | .211 | .212 | .209 | .210 |
| Const. | .211 | .211 | .212 | .209 | .210 |
| Linear | .141 | .139 | .126 | .105 | .113 |
| Boost. | .074 | .068 | .063 | .057 | .060 |
| MLP | .054 | .051 | .046 | .039 | .041 |

Table 2: Univariate Training Error Rates

Figure 4: Example Predictions on the Denoising Dataset

Figure 5: Example Predictions on the Denoising Dataset

Figure 6: Example Predictions on the Denoising Dataset

Figure 7: Example Predictions on the Horses Dataset

Figure 8: Example Predictions on the Horses Dataset

Figure 9: Example Predictions on the Horses Dataset

Figure 10: Example Predictions on the Horses Dataset

Figure 11: Example Predictions on the Horses Dataset

Figure 12: Example Predictions on the Horses Dataset