[Reviews · NeurIPS 2013]

Submitted by Assigned_Reviewer_2

Overview
========
This paper proposes an algorithm for learning general structured predictors (e.g., non-linear). This is done by replacing the structured hinge loss with its smooth dual LP relaxation and observing that optimizing over classifiers reduces to a logistic regression task. Therefore, the learning problem can be extended to cases where this optimization over the class of predictors can be solved efficiently. Specifically, the paper shows how this enables learning predictors like decision trees and multi-layer perceptrons in addition to the common linear classifiers.

Pros
====
* The observation made by the authors about reduction of the learning objective to a logistic regression problem seems novel and interesting.
* The paper is mostly clearly written and seems technically sound.
* The usefulness of the new formulation is demonstrated through experiments, and the empirical results seem convincing enough.

Cons
====
* The description of the experimental setting can probably improve.
* There seems to be some tweaking of parameters involved in the experiments (e.g., gradient descent step). It would be good to comment on the sensitivity of the results to the choice of parameters.

Comments
========
* Eq. 2: can this form include regularization?
* Alg. 1: In the linear case the objective can be optimized stochastically (as pointed out in the discussion). It may be interesting to see if/when this can be done for the other types of classifiers.
* Experiments: epsilon=1 was chosen, but it is interesting to see how the choice of different values affects the results. This is mentioned in the discussion, but seems very easy to just try empirically and report.
* Since logistic regression is solved for each factor independently, it is possible to combine different types of predictors in a single application, as shown in Table 1. It may be also interesting to optimize over several types of predictors and choose the best for each factor.
* Another relevant citation: "Efficient Training for Pairwise or Higher Order CRFs via Dual Decomposition", N. Komodakis, In CVPR 2011.

Typos
=====
* Line 36: phi should probably have subscript alpha
* Line 104: non-negativity constraint has “\geq 1” instead of “\geq 0”
* Line 158: “coordinate ascent” should be “descent” as this is a minimization problem
* Line 266: mu(y_alpha) should be mu_alpha(y_alpha)
* Eq. 8 and Line 269: x_beta should be y_beta
* Line 297: f(x,y_alpha) should be f_alpha(x,y_alpha)
Summary: This is an interesting paper which extends previous work on learning structured predictors [9, 16] to allow several non-linear predictors. The basic theoretical observation seems to be novel and valuable, and the execution meets NIPS standards.

Submitted by Assigned_Reviewer_4

The authors address the issue of learning unary and pairwise parameters for discrete valued random fields in a structured output SVM setting. In order to enable learning with non-linear function classes, the authors remove the standard l2 regularization and replace it with an entropy regularization that decomposes over the potentials of the random field. This formulation results in a saddle-point problem (typical for structured prediction) in which parameters of an LP relaxation are optimized along with parameters of the (non-linear) discriminant function. The authors use a message passing formulation, and show that with their decomposable loss, and decomposable regularizer, the problem decomposes into problems that each resemble logistic regression. Any solver for a non-linear function class can now be applied. Experimental results are given on relatively simple denoising and horse segmentation datasets showing that potentials modeled with a multi-layer perceptron performed better than other linear and non-linear methods.

Quality:
The derivation is principled and complete, with sufficient references to the optimization literature. The idea is interesting, and (given a source code release) could result in various applications of non-linear function classes in learning the parameters of graphical models. The quality of the experimental validation is somewhat weak, however, given that the datasets are quite simple, and results only compare with variants of the proposed algorithm. Plenty of citations are given for competing methods, e.g. [18] which is very closely related to the proposed approach in its goal of enabling non-linear function classes for random field models.

Clarity:
The article is well written, and clear throughout.

Originality:
[18] have previously proposed a random forest framework for learning the parameters of a random field model. This work is more general and uses a different formulation to arrive at its objective function.

Significance:
This work is quite general in the functions that can be used, though a number of limiting assumptions are made (decomposable loss, margin rescaling, only consider discrete labelings). It is an interesting approach nevertheless.

Additional comments:
Sec 3 - Limiting assumptions about the problem setting are first introduced here:
Eq (3) - only considers margin rescaling
Around line 93 - \Delta decomposes over \alpha - e.g. Hamming distance
Around line 94 - only considering discrete labelings - which is fine but not fully general in structured prediction
Decomposable loss and discrete labelings should probably already be mentioned in the introduction

Line 306 - typo in equation

Summary: An interesting approach to non-linear functions for random field models. Some limiting assumptions and somewhat weak experiments.

Submitted by Assigned_Reviewer_7

Summary:

This paper shows that parameter learning in structured-prediction problems can be reduced to independent logistic regression problems for each of the factors. This requires (a) writing an upper bound on the predictor loss via the standard margin-rescaled loss-augmented inference (LAI); (b) getting an upper bound on this upper-bound by relaxing the LAI integer programming problem to an LP; (c) smoothing the LP with a standard entropy term (still an upper-bound); (d) converting the loss-minimization from a min-max problem to a joint minimization over parameters and "messages" (lagrangian multipliers) by writing the dual of the LAI; (e) finally, using standard results about conjugate of entropy to formulate the coordinate step over parameters (given fixed messages) as a Logistic Regression training problem.

The paper shows a very nice conceptual connection between structured learning and LR. The strongest appeal (at least to me) is the ability to train non-linear structured predictors.

The only (minor) criticism of the paper is that the experiments are toy-ish and do not fully demonstrate the impact of the proposed work. It's nice to see that the errors on Weizman Horses for Boosting & MLP are lower than linear, but it would have been great to see results on more challenging datasets. Having said that, the paper contains novel high-quality work and should be accepted.

The manuscript is pretty clear, although the main novel section (Section 7) feels a little rushed and should be expanded.

Minor comments:
-- L88 is margin-scaled not slack-rescaled.
-- L101: M is typically used for the marginal polytope. Please use L for the local polytope.
-- L215: Although intuitive, but please define (Wx)_y notation before using.
-- L301: There should not be an eps on the RHS. argmax is unaffected by scaling.
Summary: Overall, a high-quality work containing a novel conceptual connection that will be broadly useful to people working in structured-prediction.
Author Feedback

Author rebuttal: Thanks to all for the helpful reviews, in which there is little to disagree with. In particular, all commented that the experiments would be improved by systematically varying epsilon, and considering larger/more realistic datasets. Such results can certainly be included in the final paper (albeit possibly in the appendix, for a lack of space).